# Equity of travel required to access first definitive surgery for liver or stomach cancer in New Zealand

Jason Gurney [1]*, Jesse Whitehead [2], Clarence Kerrison[3], James Stanley[1], Diana Sarfati[4], Jonathan Koea[5]

1 Cancer and Chronic Conditions (C3) Research Group, Department of Public Health, University of Otago, Wellington, New Zealand, 2 University of Waikato, Hamilton, New Zealand, 3 Waikato District Health Board, Auckland, New Zealand, 4 Te Aho o Te Kahu–Cancer Control Agency, Wellington, New Zealand, 5 Waitemata District Health Board, Auckland, New Zealand

* jason.gurney@otago.ac.nz

**Data Availability Statement:** The data used for this study was provided following ethical review from the New Zealand Ministry of Health National Collections team. For legal reasons, applications to

## Abstract

In New Zealand, there are known disparities between the Indigenous Māori and the majority non-Indigenous European populations in access to cancer treatment, with resulting disparities in cancer survival. There is international evidence of ethnic disparities in the distance travelled to access cancer treatment; and as such, the aim of this paper was to examine the distance and time travelled to access surgical care between Māori and European liver and stomach cancer patients. We used national-level data and Geographic Information Systems (GIS) analysis to describe the distance travelled by patients to receive their first primary surgery for liver or stomach cancer, as well as the estimated time to travel this distance by road, and the surgical volume of hospitals performing these procedures. All cases of liver (ICD-10-AM 3rd edition code: C22) and stomach (C16) cancer that occurred in New Zealand (2007–2019) were drawn from the New Zealand Cancer Registry (liver cancer: 866 Māori, 2,460 European; stomach cancer: 953 Māori, 3,192 European), and linked to national inpatient hospitalisation records to examine access to surgery. We found that Māori on average travel 120km for liver cancer surgery, compared to around 60km for Europeans, while a substantial minority of both Māori and European liver cancer patients must travel more than 200km for their first primary liver surgery, and this situation appears worse for Māori (36% vs 29%; adj. OR 1.48, 95% CI 1.09–2.01). No such disparities were observed for stomach cancer. This contrast between cancers is likely driven by the centralisation of liver cancer surgery relative to stomach cancer. In order to support Māori to access liver cancer care, we recommend that additional support is provided to Māori patients (including prospective financial support), and that efforts are made to remotely provide those clinical services that can be decentralised.

## Introduction

Liver and stomach cancers are among the most important causes of cancer death for the Indigenous Māori population of New Zealand [1]. While disparities in the key exposures that drive

access the data used in the preparation of this manuscript must be reviewed and approved by the New Zealand Ministry of Health (as custodians of these data) prior to being released to researchers. Data requests can be made by contacting data-enquiries@health.govt.nz.

**Funding:** This study was funded by the Health Research Council (HRC reference # 18/588). The funders had no role in study design, data collection and analysis, decision to publish, or preparation of the manuscript.

**Competing interests:** The authors have declared that no competing interests exist.

the incidence of these two cancers is key to the high liver and stomach cancer mortality burden affecting Māori in New Zealand [2–5], another important factor is disparities in cancer survival once diagnosed: Māori with liver cancer are nearly a third (31%) more likely to die than non-Māori liver cancer patients, and those with stomach cancer are nearly a quarter (22%) more likely to die than non-Māori stomach cancer patients [6]. Identifying and understanding the drivers of these inequities is crucial to informing systemic change aimed at dismantling them [7].

In New Zealand, there are known disparities between the Indigenous Māori and non-Māori populations in access to cancer treatment [8–15]. While the majority of both Māori and non-Māori live within cities, Māori are more likely to live outside of main centres [16], and substantially more likely to live in areas of high socioeconomic deprivation than non-Māori peoples [17]. These factors make it more difficult for Māori patients to physically access care–and since outcomes tend to be more favourable when patients are treated at high-volume facilities in metropolitan centres [18, 19], reduced capacity to travel may partially drive observed disparities in access to cancer treatment experienced by Māori. This in turn may be a partial driver of the widespread disparities in cancer survival observed between Māori and non-Māori patients [6].

There is international evidence of ethnic disparities in the distance travelled to access cancer treatment [20–23]. Much of this evidence has reported that non-White and marginalised minority groups–such as Black Americans–tend to be less likely to travel large distances when needed to access care within high-volume facilities, with this diminished access potentially an important driver of poorer cancer outcomes among these groups [20–24]. In New Zealand, patient mobility to access cancer treatment remains an under-explored area of plausible disparity between Māori and non-Māori patients.

In this paper, we use national-level data and Geographic Information Systems (GIS) analysis to describe the distance travelled by patients to receive their first primary surgery for liver or stomach cancer, as well as the estimated time to travel this distance by road, and the surgical volume of hospitals performing these procedures. We then compare these factors between Māori and European patients to consider whether there are disparities that could feasibly act as drivers of the persistent inequities in survival experienced by Māori for these cancers. Liver and stomach cancer are important causes of cancer burden for Māori, but also provide contrasting surgical contexts that are relevant for patient mobility: liver cancer surgery is highly-specialised and primarily based within high-volume hospitals in main centres; and stomach cancer surgery, while still specialised, is more routinely carried out in regional hospitals (i.e. outside the main centres).

## Methods

### Participants and data sources

All cases of liver (ICD-10-AM 3<sup>rd</sup> edition code: C22) and stomach (C16) cancer that were diagnosed in New Zealand between 2007 and 2019 were drawn from the New Zealand Cancer Registry (NZCR; liver cancer: 866 Māori, 2,460 European; stomach cancer: 953 Māori, 3,192 European; see **S1 File** for patient characteristics). These patients were linked via encrypted National Health Index number (NHI) to the National Minimum Dataset (NMDS), which contains discharge data for all publicly-funded and most privately-funded inpatient hospitalisations in New Zealand. The NMDS was used to determine access to inpatient surgeries from this same period (2007–2019), and to calculate patient comorbidity (see *Variables* below). Ethical approval was sought and received from the University of Otago Human Ethics Committee (reference # HD18/056). All data were de-identified prior to being made available by the

Ministry of Health, and our ethical approval did not require informed consent for this retrospective audit of national-level data.

## Demographic and patient variables

**Date of cancer diagnosis** was determined from the NZCR. Patient **age at diagnosis** was calculated by subtracting date of diagnosis from date of birth (also recorded on the NZCR). **Sex** (NZCR) was recorded as either female or male. Prioritised **ethnicity** was derived from the NZCR: for this study, we focus on comparisons between Māori (as the Indigenous peoples of New Zealand) and Europeans (as the majority non-Indigenous population). Patient **comorbidity** was defined using the C3 Index, a cancer-specific measure of patient comorbidity [25]. The C3 index takes all ICD-coded diagnoses recorded in the five years prior to date of diagnosis (recorded on NMDS hospital discharge records) to find 42 individual conditions, which were then weighted according to their relationship with non-cancer mortality in a cancer population [25]. Condition weights were then summed to give the final C3 score, categorised as '0' (score $< = 0$), '1' ($< = 1$), '2' ($< = 2$) and '3' ($>2$). Patients with none of the included conditions were assigned a score of 0. For our regression analysis comorbidity was included as a splined variable, using restricted cubic splines with knots placed at the 50th, 90th and 95th percentiles [26].

## First primary surgery

**Surgical procedures** were extracted from the NMDS using the Australasian College of Health Informatics (ACHI) ICD-10-AM code (3rd Edition) [27]. In order to determine a list of primary surgical procedures, clinical team members reviewed a list of all ICD-10-AM/ACHI procedures received by patients within our cohort over the study period, and determined whether a given procedure should be included. Clinical team members also identified whether a given procedure was generally undertaken with a curative or palliative intent, and also aggregated procedures into relevant groups (e.g. eight individual procedure codes pertained to partial gastrectomy). Since it was feasible that some procedures may occur in the weeks prior to diagnosis date as recorded on the NZCR, the scan included procedures that occurred up to 90 days prior to diagnosis, and up to one year post-diagnosis. NMDS data were then processed for the presence or absence of any of the included procedures for each individual patient, and the first of these procedures meeting the preceding criteria was taken as the first primary surgery.

## Patient mobility / travel

In order to compare the distance travelled to access first primary surgery, we used Geographic Information Systems (GIS) analysis to determine the distance in kilometres between the location where a given patient lived at the time of their procedure, and the location of the facility where their procedure took place. In order to do this, we derived the domicile code of patient residence [28] at the time of their procedure and the geocoded coordinates of the hospital where they underwent surgery from the NMDS [29]. Other geographic datasets used in the analysis included Beer's road network layer [30]; Statistics New Zealand's Census Area Units [31, 32], which represent the geographic units aligned with domicile codes; and the Land Information New Zealand street address dataset [33] which was used to create address-weighted centroids for each domicile.

   **Distance and travel times to surgery** for Māori and non-Māori patients were estimated using the OD-Matrix function within the GIS software ArcGIS (Environmental Systems Research Institute, U.S.A.). To do this, we estimated the road network distance and travel time between the address-weighted centroid of each patient's domicile of residence, and the

geocoded coordinates of each hospital where a surgery took place. Missing data prevented the attribution of distance or travel time for 104 patients (1%) who underwent surgery during the study period.

One-way distance from domicile of residence to location of treatment by road was expressed in kilometres (km), and categorised as follows: <25km, 25-99km, 100-199km, and >200km. These categories were chosen to approximately represent a range of typical travelling patterns, including close or across-town travel (<25km), travel from surrounding districts (25-99km), regional travel from nearby cities or towns (100-199km) and inter-regional travel (>200km). One-way travel time was expressed in minutes (mins), and categorised as follows: <60mins, 60-149mins, and >150mins. These travel times were chosen to approximately represent close, across-town or surrounding district travel (<60mins), regional travel from nearby cities or towns (60-149mins, or 1–2.5 hours), and inter-regional travel (>150mins, or more than 2.5 hours).

## Hospital volume

In order to investigate whether Māori and European patients differ in terms of access to high-volume surgical facilities, we focussed on four key procedures (two curative, two palliative) that were found to occur relatively commonly among our cohort. For liver cancer, we examined minor hepatectomy and liver ablation; while for stomach cancer, we examined partial gastrectomy and endoscopic injection.

After this, we extracted data from the NMDS for every instance where one of these procedures was conducted over the study period (i.e. in the total population, not just the cohort used for this study). We then used facility codes from the NMDS to determine the facility where each given procedure took place (e.g. Auckland City Hospital). We then created frequency tables for each individual procedure, which showed how often each procedure was undertaken at each given facility (**S1 File**). Based on these tables, we categorised the volume of each facility for each individual procedure. To do this, we scanned each table to determine the placement of clear thresholds in terms of procedures undertaken per year, and set volume categories accordingly: for example, for minor hepatectomies, the data showed clear breaks between those facilities that undertook one minor hepatectomy per week (high volume), one every 1–2 months (medium volume), and less than or equal to one per year (low volume). A full list of these frequency tables, their volume category and a rationale for selection of this category is included in **S1 File**.

## Statistical analysis

For our **descriptive analysis**, we determined frequencies and both crude (i.e. unadjusted) and age-standardised proportions, stratified by cancer type and ethnicity. **Age-standardised proportions**, were determined using direct standardisation methods [34], with the total Māori cancer population 2007–2019 (30,346) as the standard population [35, 36]. We conducted three separate sets of **logistic regression models** in order to compare outcomes between Māori and European patients. We compared Māori and European patients in terms of the likelihood of both a) belonging to a given travel distance category (e.g. <25 kilometres); b) belonging to a given time category (e.g. <60 minutes), or c) belonging to a given hospital volume category (e.g. high-volume). To do this, we calculated crude and adjusted logistic regression models, by cancer type, with European patients as the reference group. Odds ratios (OR) and their 95% confidence intervals were extracted from crude models, as well as models that adjusted for the confounding impact of age, sex, the type of first primary surgery (e.g. minor hepatectomy for liver cancer patients) and comorbidity. Age and sex were included in the

adjusted models as classic confounders; procedure type was included in order to control for possible differences in the types of procedures undertaken on Māori vs. European patients, wherein Māori could feasibly be more likely to undergo procedures that require treatment in main centres; and comorbidity was included in order to control for patient morbidity at the time of treatment, wherein due to increased morbidity load Māori could feasibly be more likely to undergo procedures in main centres with the capacity to meet more complex care needs.

## Results

### Type of surgery

Frequencies and proportions (crude and age-standardised) of first primary curative or palliative surgery for liver and stomach cancer are shown in **Table 1**, stratified by ethnicity. For liver cancer, we found that more than half of the first primary surgeries accessed by both Māori and European patients were palliative liver ablation procedures (age-standardised proportions: Māori 57%, European 56%). Nearly a quarter (23%) of Māori liver cancer patients underwent a minor hepatectomy as their first primary surgery (compared to 11% of Europeans), while only 3% of Māori (compared to 6% of Europeans) underwent a liver transplant.

For stomach cancer, nearly half of Māori patients underwent a partial gastrectomy (47%), compared to around a quarter of European patients (26%), while nearly a third of Māori

**Table 1. Frequencies and proportions (crude and age-standardised) of first primary curative or palliative surgery for liver and stomach cancer, stratified by ethnicity.**

| Cancer | Surgery Type | Māori | | | European | | |
|---|---|---|---|---|---|---|---|
| | | n | % | Age Std. % | n | % | Age Std. % |
| *Liver* | Total Primary Surgeries | 288 | - | - | 668 | - | - |
| | *Curative Surgeries* | | | | | | |
| | Major Hepatectomy | 30 | 10% | 11% | 61 | 9% | 11% |
| | Minor Hepatectomy | 67 | 23% | 23% | 107 | 16% | 16% |
| | Percutaneous Drainage | 6 | 2% | 2% | 7 | 1% | 2% |
| | PTC | 5 | 2% | 2% | 21 | 3% | 3% |
| | Transplant | 9 | 3% | 3% | 38 | 6% | 6% |
| | *Palliative Surgeries* | | | | | | |
| | Endoscopic Injection | 6 | 2% | 2% | 20 | 3% | 3% |
| | Hepaticoenterostomy | 0 | 0% | 0% | 3 | 0% | 0% |
| | Liver Ablation | 164 | 57% | 57% | 408 | 61% | 56% |
| | TIPS | 1 | 0% | 0% | 3 | 0% | 0% |
| *Stomach* | Total Primary Surgeries | 377 | - | | 969 | - | |
| | *Curative Surgeries* | | | | | | |
| | Oesophagectomy | 17 | 5% | 4% | 281 | 29% | 33% |
| | Partial Gastrectomy | 173 | 46% | 47% | 290 | 30% | 26% |
| | Percutanoeus Drainage | 1 | 0% | 0% | 3 | 0% | 0% |
| | Total Gastrectomy | 136 | 36% | 32% | 257 | 27% | 26% |
| | *Palliative Surgeries* | | | | | | |
| | Pyloroplasty | 1 | 0% | 0% | 1 | 0% | 0% |
| | Endoscopic Injection | 42 | 11% | 12% | 127 | 13% | 11% |
| | Enteroenterostomy | 5 | 1% | 1% | 10 | 1% | 1% |
| | Tumour Debulking | 2 | 1% | 0% | 0 | 0% | 0% |

PTC: Percutaneous transhepatic cholangiography; TIPS: transjugular intrahepatic portosystemic shunt.

underwent a total gastrectomy (32%), compared to 26% of European patients. Europeans were much more likely to undergo an oesophagectomy (Māori 4%, European 33%). A similar proportion of Māori and European patients underwent palliative endoscopic injections as their first primary surgical treatment (Māori 12%, European 11%; **Table 1**).

## Patient mobility

Frequencies and proportions of one-way distance travelled and travel time to first primary surgery for Māori and European patients, along with crude and adjusted odds ratios, are shown in **Table 2**. For liver cancer, we found that Māori were marginally less likely to live less than 25 kilometres from the location of their first primary surgery than Europeans, although the odds ratio confidence intervals crossed the null (Māori 37%, European 42%, age, sex, comorbidity and procedure type adjusted odds ratio [adj. OR] 0.80, 95% CI 0.59–1.07), and more likely to live 200+ kilometres away (Māori 36%, European 29%; adj. OR 1.48, 95% CI 1.09–2.01). The median travel distance for Māori was approximately twice that of European patients (Māori 121km, European 56km), although the interquartile ranges (IQR) were wide for both groups (Māori 14-287km, European 11-221km). Reflected in these differences in distance to first surgery were differences in travel time: Māori were somewhat less likely to live either less than an hour away (Māori 45%, European 51%; adj. OR 0.78, 95% CI 0.58–1.05) or between an hour and 2.5 hours away (Māori 10%, European 14%; adj. OR 0.69, 95% CI 0.44–1.08), and considerably more likely to live at least 2.5 hours away (Māori 44%, European 33%; adj. OR 1.55, 95%

**Table 2. Patient mobility to first primary surgery, in one-way distance and travel time, for Māori and non-Māori liver and stomach cancer patients.**

| Cancer | Patient Mobility | Māori | | | European | | | Odds Ratios (95% CI) | |
|---|---|---|---|---|---|---|---|---|---|
| | | *n* | % | *Age Std. %* | *n* | % | *Age Std. %* | *Crude* | *Adjusted* |
| *Liver* | Distance to First Primary Surgery (Km): | | | | | | | | |
| | <25 Kilometres | 104 | 36% | 37% | 273 | 41% | 42% | 0.82 (0.62–1.09) | 0.8 (0.59–1.07) |
| | 25–100 Kilometres | 32 | 11% | 11% | 84 | 13% | 12% | 0.87 (0.57–1.34) | 0.91 (0.58–1.44) |
| | 100–200 Kilometres | 42 | 15% | 14% | 110 | 17% | 15% | 0.87 (0.59–1.28) | 0.85 (0.57–1.26) |
| | 200+ Kilometres | 109 | 38% | 36% | 196 | 30% | 29% | 1.47 (1.1–1.96) | 1.48 (1.09–2.01) |
| | Median in Kilometres (IQR) | 121 km (14–287) | | | 56 km (11–221) | | | | |
| | Travel Time to First Primary Surgery (Mins) | | | | | | | | |
| | <60 minutes | 127 | 44% | 45% | 334 | 50% | 51% | 0.79 (0.6–1.05) | 0.78 (0.58–1.05) |
| | 60–150 minutes | 31 | 11% | 10% | 100 | 15% | 14% | 0.69 (0.45–1.06) | 0.69 (0.44–1.08) |
| | 150+ minutes | 129 | 45% | 44% | 229 | 35% | 33% | 1.56 (1.18–2.06) | 1.55 (1.15–2.08) |
| | Median in Minutes (IQR) | 123 mins (22–252) | | | 59 mins (19–204) | | | | |
| *Stomach* | Distance to First Primary Surgery (Km): | | | | | | | | |
| | <25 Kilometres | 196 | 52% | 51% | 512 | 56% | 51% | 0.96 (0.76–1.21) | 0.88 (0.67–1.16) |
| | 25–100 Kilometres | 89 | 24% | 23% | 227 | 25% | 22% | 1 (0.76–1.33) | 1.09 (0.79–1.5) |
| | 100–200 Kilometres | 61 | 16% | 16% | 95 | 10% | 9% | 1.76 (1.25–2.49) | 1.98 (1.31–2.98) |
| | 200+ Kilometres | 29 | 8% | 7% | 87 | 9% | 10% | 0.84 (0.54–1.3) | 0.92 (0.54–1.57) |
| | Median in Kilometres (IQR) | 22 km (7–96) | | | 21 km (7–74) | | | | |
| | Travel Time to First Primary Surgery (Mins): | | | | | | | | |
| | <60 minutes | 256 | 68% | 67% | 674 | 73% | 67% | 0.91 (0.71–1.17) | 0.76 (0.57–1.03) |
| | 60–150 minutes | 88 | 23% | 23% | 150 | 16% | 15% | 1.65 (1.23–2.21) | 2.11 (1.48–2.99) |
| | 150+ minutes | 31 | 8% | 8% | 97 | 11% | 11% | 0.8 (0.53–1.22) | 0.96 (0.57–1.6) |
| | Median in Minutes (IQR) | 28 mins (12–82) | | | 26 mins (13–68) | | | | |

Adjusted model adjusts for age, sex, type of first primary surgery, and comorbidity.

CI 1.15–2.08). The median travel time for Māori was more than two hours (123mins) compared to less than hour for Europeans (59mins), although the IQR was again wide for both groups (Māori 22-252mins, European 19-204mins).

For stomach cancer, there was no apparent difference between Māori and European patients in the proportion of patients living less than 25km away (both 51%; OR 0.88, 95% CI 0.67–1.16) or 25-100km away (Māori 23%, European 22%; adj. OR 1.09, 95% CI 0.79–1.50); however, Māori were more likely to live 100-200km away (Māori 16%, European 9%; adj. OR 1.98, 95% CI 1.31–2.98) and similarly likely to live 200km+, although the confidence limits were wide (Māori 7%, European 10%; adj. OR 0.92, 95% CI 0.54–1.57). The median distances were largely equivalent between the two groups (Māori 22km, European 21km), although IQRs were again wide (Māori 7-96km, European 7-74km). These observations were reflected in terms of travel time: Māori appeared more likely to travel between one hour and 2.5 hours (Māori 23%, European 15%; adj. OR 2.11, 95% CI 1.48–2.99), while marginally less likely to travel less than one hour (both 67%, but adj. OR 0.76, 95% CI 0.57–1.03) or at least 2.5 hours (Māori 8%, European 11%; adj. OR 0.96, 95% CI 0.57–1.60). There was no obvious difference in median travel time between groups (Māori 28mins, European 13–68 minutes), again with wide IQRs (Māori 12-82km, European 13-68km; Table 2).

## Hospital volume

A comparison of the place of surgery according to hospital volume between Māori and European patients is shown in Table 3, for the previously selected relatively common procedures. We found that more Māori liver cancer patients who underwent a minor hepatectomy did so in a high-volume hospital compared to European patients (Māori 76%, European 53%; adj. OR 3.36, 95% CI 1.60–7.04), and were commensurately less likely to undergo the procedure in a medium-volume hospital (Māori 15%, European 41%; adj. OR 0.30, 95% CI 0.14–0.64).

**Table 3. Comparison of place of surgery according to hospital volume between Māori and European patients for selected surgeries.**

| Cancer | Hospital Volume | Māori | | | European | | | Odds Ratios (95% CI) | |
|---|---|---|---|---|---|---|---|---|---|
| | | n | % | Age Std. % | n | % | Age Std. % | Crude | Adjusted |
| *Liver* | *Minor Hepatectomy* | | | | | | | | |
| | High Volume | 54 | 83% | 76% | 55 | 54% | 53% | 3.93 (1.92–8.02) | 3.36 (1.6–7.04) |
| | Medium Volume | 11 | 17% | 15% | 45 | 45% | 41% | 0.27 (0.13–0.57) | 0.3 (0.14–0.64) |
| | Low Volume | 0 | 0% | 0% | 1 | 1% | 1% | - | - |
| | *Liver Ablation* | | | | | | | | |
| | High Volume | 128 | 78% | 77% | 307 | 75% | 75% | 1.17 (0.76–1.8) | 0.99 (0.63–1.56) |
| | Medium-High Volume | 14 | 9% | 9% | 37 | 9% | 10% | 0.94 (0.49–1.78) | 0.98 (0.51–1.92) |
| | Medium-Low Volume | 20 | 12% | 13% | 54 | 13% | 12% | 0.91 (0.53–1.58) | 1.06 (0.6–1.89) |
| | Low Volume | 2 | 1% | 2% | 10 | 2% | 3% | 0.49 (0.11–2.27) | 0.85 (0.17–4.2) |
| *Stomach* | *Partial Gastrectomy* | | | | | | | | |
| | High Volume | 85 | 50% | 50% | 154 | 56% | 51% | 0.81 (0.55–1.18) | 0.78 (0.51–1.17) |
| | Medium-High Volume | 58 | 34% | 34% | 85 | 31% | 32% | 1.18 (0.79–1.77) | 1.16 (0.75–1.81) |
| | Medium-Low Volume | 24 | 14% | 14% | 32 | 12% | 8% | 1.67 (0.92–3.02) | 1.75 (0.92–3.35) |
| | Low Volume | 2 | 1% | 1% | 3 | 1% | 1% | 1.09 (0.18–6.62) | 2.19 (0.31–15.62) |
| | *Endoscopic Injection* | | | | | | | | |
| | High Volume | 14 | 34% | 36% | 55 | 47% | 41% | 0.66 (0.32–1.36) | 0.67 (0.31–1.47) |
| | Medium Volume | 20 | 49% | 44% | 38 | 32% | 26% | 2.13 (1.04–4.35) | 2.57 (1.17–5.64) |
| | Low Volume | 7 | 17% | 18% | 25 | 21% | 26% | 0.82 (0.33–2.05) | 0.57 (0.21–1.54) |

Adjusted model adjusts for age, sex, type of first primary surgery, and comorbidity.

There were no apparent differences between Māori and European patients in terms of the volume of liver ablation procedures, with the majority of both groups undergoing this procedure in a high-volume hospital (Māori 77%, European 75%; adj. OR 0.99, 95% CI 0.63–1.56).

There was also no strong evidence for a difference between Māori and European stomach cancer patients in terms of partial gastrectomy procedures, with both groups undergoing this procedure in either a high-volume (Māori 50%, European 51%; adj. OR 0.78, 95% CI 0.51–1.17) or medium-high-volume facility (Māori 34%, European 32%; adj. OR 1.16, 95% CI 0.75–1.81). Māori appeared more likely to access endoscopic injections in medium-volume hospitals (Māori 44%, European 26%; adj. OR 2.57, 95% CI 1.17–5.64) and commensurately less likely to access high- and low-volume hospitals, but low numbers of procedures affected the precision of this finding (**Table 3**).

## Discussion

We found that Māori with liver cancer are less likely to live close to treatment, and thus need to travel much further than European patients to access first primary surgery–even after adjusting for the type of treatment they are undergoing. Māori, on average, travel 120km for liver cancer surgery, compared to around 60km for Europeans. A substantial minority of both Māori and European liver cancer patients must travel more than 200km for their first primary liver surgery, and this situation appears worse for Māori (36% vs 29%; adj. OR 1.48, 95% CI 1.09–2.01). These disparities correspond to substantial differences in estimated travel time, with Māori estimated to spend more than two hours travelling to their first surgery, compared to less than one hour for Europeans (**Table 2**).

Even after adjusting for factors including comorbidity, Māori with liver cancer who received a minor hepatectomy were considerably more likely to undergo their procedure in a high-volume hospital. This likely reflects the volume of Māori treated at Auckland City Hospital, which was the sole high-volume hospital for this procedure (performing approximately one/week over the study period, **S1 File**). This in-turn may be a key driver of the substantial differences in travel distances and times observed between Māori and European patients; in other words, it may be that Māori were more likely to need to travel from outside Auckland to this main centre in order to receive treatment.

For stomach cancer, there were clear differences between Māori and non-Māori in relation to the type of procedure received, with Māori patients being more likely to be treated with partial gastrectomy than non-Māori patients (47% vs 26%). This is likely driven by the relatively higher proportion of distal stomach cancer among Māori, which we have described previously [13]. However, in contrast to liver cancer, differences between ethnic groups in patient mobility were less stark for stomach cancer. We found that around 75% of both Māori and European patients lived within 100km of the location of their first primary surgical treatment, and there was no difference in median distances or travel times. Māori were perhaps more likely to live 1–2.5 hours away from treatment; however, these differences are unlikely to be meaningful in the context of the broader trends observed for this cancer for two reasons: 1) because it only applies to a minority of Māori and European patients; and 2) because Māori were also somewhat less likely to live more than 2.5 hours away–meaning that Māori being more likely to live 1–2.5 hours away did not represent a trend toward Māori living further from stomach cancer treatment than European patients. The fact that the majority of both Māori and European patients lived relatively close to the location of their first surgery is likely driven by the greater regional delivery of surgical care for this cancer compared to liver cancer. The tables in **S1 File** show a greater number of high- or medium-volume facilities providing surgical care for stomach cancer than those providing surgical care for liver cancer.

## What do our findings mean?

Our findings in the liver cancer context make it clear that Māori must travel further–and spend more time travelling–than European patients in order to access surgical care. This has multiple ramifications.

- Firstly, there are the direct costs (financial and otherwise) in requiring Māori to travel further to access their care. These include a) greater costs of transport to access care (e.g. petrol costs, parking, public transport costs); b) greater accommodation and sustenance costs for those travelling from out of town (e.g. from outside Auckland); c) greater need for time away from work, and associated potential loss of income; and d) greater need for time away from home and whānau (family), which may require Māori to alter childcare or elderly care arrangements, placing greater burden on whānau to cover these support arrangements. These direct consequences of the greater burden of travel on Māori patients–many of which are financial–must be viewed alongside substantial disparities in socioeconomic stability [17]; in other words, Māori need to travel further for care, but inversely are less likely to be able to cover the costs of this travel. This disparate financial burden is compounded by the way in which financial support (through schemes like the National Travel Assistance Scheme, or NTA) are paid out, which often requires up-front coverage of costs by patients and their whānau, with subsequent reimbursement.

- Secondly, there are the indirect costs of disparities in required travel experienced by Māori. These include a) the greater impact of travel on job stability for patients (and their support person, if applicable), who may be required to take extended leaves of absence from their workplace in order to access treatment compared to care provided closer to home; b) the greater impact of travel on whānau and friends to fill support 'gaps' while the patient (and their support person) are absent from home; and c) the greater absence of Māori patients from their homes and communities means they will be less available to support their own whānau and friends, participate in community activities, or similar.

- Finally, there are ramifications relating to care access: while our analysis was limited to those who received surgery, it remains unclear the extent to which the differential travel burden experienced by Māori liver cancer patients acts as a barrier to receiving treatment in the first place. While it is not possible to examine this with the data available for this study, it is conceivable that for some Māori patients, the challenges presented by the above factors are too great to overcome, and so these patients cannot access care. However, we note that our previous clinical audits have not observed strong disparities in access to surgical care for liver and stomach cancer [5, 13]–suggesting that in spite of facing greater barriers to accessing surgery, Māori patients are finding ways to overcome these challenges, including by travelling to high-volume centres for their care. This somewhat contrasts with observations from the US noted in the Introduction [20–24], although the two contexts are unlikely to be directly comparable.

The general absence of travel disparities in the stomach cancer context provides a key piece of evidence: it tell us that the travel disparities observed in liver cancer are not transferable across cancer contexts, even within the context of upper-gastrointestinal cancers. Rather, they are likely to be the consequence of the centralisation of liver cancer surgical care to a few treatment hubs in main metropolitan areas–as opposed to stomach cancer, where surgical care is more widely available across regions. This diffuse model of care brings care closer to the communities from which patients arise, and given the above disparities observed in liver cancer, care that is delivered closer to home will disproportionately benefit Māori. As such, spatially-

equitable distributions of health services, which provide better access to populations with greater health needs, have the potential to contribute to improved health equity.

However, avoiding disparities in travel burden via the decentralisation of liver cancer surgical care is not straightforward. The surgical treatment of liver cancer requires high-volume centres and specialised surgeons [18, 19]. In other international contexts, including Europe and North America, complex cancer surgeries such as liver surgeries are being increasingly centralised to specialised high-volume centres, in response to growing evidence of better patient outcomes in high-volume versus low-volume centres [37]. New Zealand is sparsely populated and diffuse compared to countries in these regions, making it difficult to achieve high surgical volumes in a given facility for a given procedure. As noted earlier, even our highest-volume hospital performs the most common primary liver cancer surgery (minor hepatectomy) only once per week. Given these constraints, further decentralisation of these complex surgeries would likely result in poorer outcomes for patients. There are other complex cancer surgeries that are similarly constrained in terms of requiring centralisation to ensure optimal outcomes, and for which the same issues are likely to apply.

So how do we reduce the disparate travel burden on our Māori liver cancer patients? We believe there are some things that can be done in the short- to medium-term to achieve this.

- Firstly, schemes such as the NTA should provide patients and/or their whānau with up-front funding (rather than relying on reimbursement systems), with this funding to be set at an appropriate level that recognises both the direct and indirect costs associated with travelling for cancer care. We also need to ensure that whānau are aware of what funding is available and understand how to access it [38].

- Secondly, to the extent that is possible, clinics requiring attendance in the lead-up to and following surgery should be held closer to home for the patient. We need more opportunities for the gastroenterology workforce to move between hospitals, advancing care for hepatitis, cirrhosis and liver cancer patients in regions that are underserved in this respect. We need to support non-metropolitan centres with diagnostics, and facilitate collaborative approaches to care in the regions including involvement in multidisciplinary team meetings (MDMs). By way of example, the clinical pathway for liver transplant candidates typically involves multiple investigations, some of which are not available in regional centres, followed by trips to the liver transplant unit at Auckland City Hospital for further investigations (including psychological and social assessments), followed by ongoing assessment every few months by hepatologists in the lead-up to transplant. While in this case the surgery itself needs to be centralised in a high-volume centre, many of these assessments could be conducted remotely, reducing the burden on the patient–which would again disproportionately benefit Māori. In short, rather than the onus of travel always resting with the patient, wherever possible we should be bringing the care to them.

- Thirdly, there are other ways that we could support Māori liver cancer patients, and those requiring complex care, as they navigate the need to travel further for their care than other patients. For example, we should recognise that for some whānau, more than one support person will be needed to accompany the patient and provide auxiliary care, and consideration should be given to funding these additional support people to perform this service. There is also a need for more Māori cancer care coordinators or other Māori health support workers to help patients navigate their care and cushion the complexity of their various appointments, including helping the patient to understand their options and demystify clinical information. Finally, there is a need for more resources for whānau to help them understand the pathway of liver cancer treatment, due to its complexity; such resources could

include accounts (e.g. videos) of Māori patients who have previously navigated the liver cancer pathway.

In summary, our findings suggest that the centralisation of liver cancer treatment results in strong disparities in required travel between Māori and European patients; however, given the complexity of these procedures, coupled with the current volume of procedures, decentralisation may not be a viable manner of addressing this disparate burden. We have outlined above multiple mechanisms by which this burden might be mitigated for Māori patients, and believe these (and other) initiatives aimed at reducing barriers to care for Māori should be prioritised during the health system reforms that are currently occurring in New Zealand.

## Strengths and limitations

A strength of the current study is the national coverage of our dataset: we have examined all cases of liver and stomach cancer on the New Zealand Cancer Registry, and analysed access to treatment using national data. We also note that this is among the first studies to examine disparities in patient mobility to access cancer treatment conducted in New Zealand. In terms of limitations, we note that we have measured travel distance and estimated drive time in terms of travel by road, whereas travel from regions that are widely separated (e.g. Invercargill to Auckland) is likely to have occurred by plane. However, the key theme–that these patients need to travel further, with the consequent ramifications–remains in spite of this caveat. In our regression models, we adjusted for procedure type to control for the possibility that Māori may be more (or less) likely to get some procedures than European patients (and those procedures may be more likely to be performed in central hubs). This is sensible in situations where the explanation for Māori receiving a procedure relates to differential disease (such as the higher burden of distal stomach cancer); however, the situation grows more complex when we consider explanations that are related to travel distance itself (e.g. Māori accessing some procedures more than others because they are less likely to be able to travel). It is possible that this effort to control for known differences in the types of procedures accessed by Māori patients (**Table 1**) has potentially over-adjusted for the impact of procedure type.

We have only included data on surgery in this manuscript: future work aiming to understand the impact of travel burden on care access should also include data on radiotherapy and systemic therapy, which by their nature will likely require repeated travel. As such the current study does not provide a comprehensive picture of the total travel burden, nor disparities in such burden between ethnic groups, but rather a snapshot representation of that burden. Linked to this, we have not described the economic ramifications of differences in required travel between ethnic groups; this would be a useful future research direction that would build directly on the findings of the current study.

Finally, we have focussed on travel required to access cancer treatment within this manuscript; however, in the broader context of addressing disparities in cancer survival for Māori patients, it is also important to examine travel required to access cancer diagnosis. Such an analysis is out of scope for the current study, but future work in this area could include an assessment of ethnic differences in travel distance to screening hubs and secondary diagnostic facilities.

## Conclusions

In our examination of patient mobility to access first primary surgery for liver and stomach cancer, we found that Māori liver cancer patients needed to travel further, for longer, in order to access care. No such disparities were observed for stomach cancer. This contrast between

procedures is likely driven by the centralisation of liver cancer surgery relative to stomach cancer, with this centralisation of care having direct and indirect implications that disproportionately impact Māori. In order to support Māori to access liver cancer care in spite of these disparities, we recommend that financial support is provided prospectively and at a level that is in congruence with need; that new and renewed efforts are made to provide clinical services that can be decentralised and provided remotely; and that Māori liver cancer patients are provided with further support on their care journey via funding of additional support people, provision of Māori cancer care coordinators and other Māori health support staff, and the development of further resources that enable and facilitate Māori understanding of their liver cancer journey.

## Supporting information

**S1 File. The Supplementary Material that accompanies this manuscript includes demographic and patient characteristics of all Māori and European patients diagnosed with liver and stomach cancer between 2017–2019, and hospital volume categorisation for each of the four procedures included in this study.**
(DOCX)

## Acknowledgments

We would like to thank Chris Lewis at the Ministry of Health for providing the National Collections data extract for this study.

## Author Contributions

**Conceptualization:** Jason Gurney, Jonathan Koea.

**Data curation:** Jason Gurney, Jesse Whitehead.

**Formal analysis:** Jason Gurney, Jesse Whitehead.

**Funding acquisition:** Jason Gurney.

**Investigation:** Jason Gurney, Jesse Whitehead, Clarence Kerrison, James Stanley, Diana Sarfati, Jonathan Koea.

**Methodology:** Jason Gurney, James Stanley, Diana Sarfati, Jonathan Koea.

**Project administration:** Jason Gurney.

**Writing – original draft:** Jason Gurney.

**Writing – review & editing:** Jason Gurney, Jesse Whitehead, Clarence Kerrison, James Stanley, Diana Sarfati, Jonathan Koea.

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
