## [Decision Letter · Decision Letter 0]

30 Dec 2021

PONE-D-21-38090Equity of travel required to access first definitive surgery for liver or stomach cancer in New ZealandPLOS ONE

Dear Dr. Gurney,

Thank you for submitting your manuscript to PLOS ONE. After careful consideration, we feel that it has merit but does not fully meet PLOS ONE’s publication criteria as it currently stands. Therefore, we invite you to submit a revised version of the manuscript that addresses the points raised during the review process.

We look forward to receiving your revised manuscript.

Kind regards,

Aloysious Dominic Aravinthan, MBBS, FRCP, PhD

Academic Editor

PLOS ONE

Journal Requirements:

This study was funded by the Health Research Council (HRC reference # 18/588).  We would like to thank Chris Lewis at the Ministry of Health for providing the National Collections data extract for this study.

No - the funders had no role in study design, data collection and analysis, decision to publish, or preparation of the manuscript.

Additional Editor Comments:

This is a well written and presented study and the authors should be congratulated for their effort. however, as the reviewers point out there are number of limitations that needed to be addressed before further consideration for publication.

Reviewers' comments:

Reviewer's Responses to Questions

**Comments to the Author**

1. Is the manuscript technically sound, and do the data support the conclusions?

Reviewer #1: No

Reviewer #2: Yes

2. Has the statistical analysis been performed appropriately and rigorously? 

Reviewer #1: No

Reviewer #2: Yes

3. Have the authors made all data underlying the findings in their manuscript fully available?

Reviewer #1: Yes

Reviewer #2: Yes

4. Is the manuscript presented in an intelligible fashion and written in standard English?

Reviewer #1: Yes

Reviewer #2: Yes

5. Review Comments to the Author

Reviewer #1: The aim of studying disparities between different populations in the accessibility to medical treatments is very important. Distance and travel time can be major contributors for such disparities.

There are some concerns about the methodology of this study:

1. The differences between the populations can be related to other factors such as the incidence compare to general population, stage of the disease in the diagnosis and prior treatment before surgery (neoadjuvant /local)

2. In the case of liver cancer there also influence of the primary liver disease and other comorbidities on the decision about the treatment.

3. There was no explanation about the differences between Maori and European in stomach cancer when the distance was 100-200km but not for >200km and for travel time of 60-150min but not for >150min.

4. The conclusion of the study is that there are differences in distance and travel time , there is need to show if this differences link to different accessibility to treatment (there were no statistical comparison in the rate of different surgery type between the groups and normalize it to the rate in general population).

Reviewer #2: This is a straight forward paper describing the difference in time/distance for Maori patients to attend a liver cancer centre compared to patients of European descent. This would be important to highlight the resulting disparity in economic and logistic burden this creates. Whilst the distance and time is described, it might help the reader to quantify this burden better if the authors also described difference in costs likely to be entailed.

The statistics seemed fine and the authors have carefully calculated the distances and times of travel appropriately.

An important part of the problem described though is whether these differences impact outcomes. It might be that there is no difference in stage or outcome as a consequence of these disparities in distance. Centralising the services might mean that better liver cancer care is being provided at high volume specialist centres. More important than local access to curative surgery might be access to diagnostic services to allow early diagnosis of cancer. Therefore staging at time of diagnosis would be important to at least discuss. If the author's can't show this from their data is there data from elsewhere? They mention that Maori patients had more co-morbidity which they used to adjust the disparities in high/low volume centre use, but the underlying co-morbidity data isn't presented and should be in table 1 if it is included in the later models.

Similarly it would be helpful to discuss if there is any difference in outcomes of liver cancer between the patient groups and if that is associated with the disparities in distance / timing? Presumably even if it cannot be included at patient level in this study there is national cancer outcome data with ethnicity to inform the discussion? I may have missed it but I couldn't see if this was mentioned.

For the multivariate models, these should be available in tables with all their included covariates. If not in the main paper these should be in the supplementary information. The tables with the adjusted odds ratios should at least list the included covariates in the table footnotes.

6. PLOS authors have the option to publish the peer review history of their article (what does this mean?). If published, this will include your full peer review and any attached files.

Reviewer #1: No

Reviewer #2: No

---

## [Author Response · Author response to Decision Letter 0]

23 Feb 2022

Equity of travel required to access first definitive surgery for liver or stomach cancer in New Zealand

Response to Reviewers Document

We would like to thank the Reviewers and Editors for the time taken to consider our manuscript. We have responded to each Reviewer comment below.

Reviewer One

The aim of studying disparities between different populations in the accessibility to medical treatments is very important. Distance and travel time can be major contributors for such disparities.

There are some concerns about the methodology of this study:

1. The differences between the populations can be related to other factors such as the incidence compare to general population, stage of the disease in the diagnosis and prior treatment before surgery (neoadjuvant /local)

Thank you for this comment. The Reviewer may be suggesting that perhaps the differences in distance and time travelled to access treatment might be related to factors that we have not controlled for in the study. In terms of background incidence, this cannot conceivably impact our results since all patients within the study had either liver or stomach cancer; we cannot think of a way in which differences in background incidence might impact travel among those already diagnosed. In terms of stage of disease at diagnosis, we have previously shown that there are no meaningful differences in stage at diagnosis for liver or stomach cancer between Māori and non-Māori patients1-3 and this can also be seen in Supplementary Table 1. In terms of accessing prior treatment, if there were differences between ethnic groups in this access, this would not alter the primary focus of the study (i.e. to measure distance travelled to first primary surgical treatment).

2. In the case of liver cancer there also influence of the primary liver disease and other comorbidities on the decision about the treatment.

Thank you – we have adjusted for comorbidity within our results, and discuss these adjusted results within our Results and Discussion sections.

3. There was no explanation about the differences between Maori and European in stomach cancer when the distance was 100-200km but not for >200km and for travel time of 60-150min but not for >150min.

Thank you for this comment. We have updated the relevant part of the Discussion section with the following text:

‘Māori were perhaps more likely to live 1-2.5 hours away from treatment; however, these differences are unlikely to be meaningful in the context of the broader trends observed for this cancer for two reasons: 1) because it only applies to a minority of Māori and European patients; and 2) because Māori were also somewhat less likely to live more than 2.5 hours away – meaning that Māori being more likely to live 1-2.5 hours away did not represent a trend toward Māori living further from stomach cancer treatment than European patients’

4. The conclusion of the study is that there are differences in distance and travel time, there is need to show if this differences link to different accessibility to treatment (there were no statistical comparison in the rate of different surgery type between the groups and normalize it to the rate in general population).

Thank you for this comment. We have shown the crude and adjusted rates (and rate ratios) of treatment access within Table 2 of the manuscript for all patients diagnosed with liver and stomach cancer within the New Zealand population over the study period (in other words, the data presented in this study include all cancer surgeries performed on patients diagnosed with these cancers over this period, ensuring their generalisability).

 

Reviewer Two

This is a straight forward paper describing the difference in time/distance for Maori patients to attend a liver cancer centre compared to patients of European descent. This would be important to highlight the resulting disparity in economic and logistic burden this creates. Whilst the distance and time is described, it might help the reader to quantify this burden better if the authors also described difference in costs likely to be entailed.

Thank you for this comment. Understanding the economic ramifications of the observed differences would be useful and informative, but due to the volume of work required to complete the current analysis, this is outside of the scope of the current manuscript. We have added the following note to the Limitations section of the manuscript to acknowledge this:

‘We have only included data on surgery in this manuscript: future work aiming to understand the impact of travel burden on care access should also include data on radiotherapy and systemic therapy, which by their nature will likely require repeated travel. As such the current study does not provide a comprehensive picture of the total travel burden, nor disparities in such burden between ethnic groups, but rather a snapshot representation of that burden. Linked to this, we have not described the economic ramifications of differences in required travel between ethnic groups; this would be a useful future research direction that would build directly on the findings of the current study.’

The statistics seemed fine and the authors have carefully calculated the distances and times of travel appropriately.

Thank you for this comment.

An important part of the problem described though is whether these differences impact outcomes. It might be that there is no difference in stage or outcome as a consequence of these disparities in distance. Centralising the services might mean that better liver cancer care is being provided at high volume specialist centres. More important than local access to curative surgery might be access to diagnostic services to allow early diagnosis of cancer. Therefore staging at time of diagnosis would be important to at least discuss. If the author's can't show this from their data is there data from elsewhere? They mention that Maori patients had more co-morbidity which they used to adjust the disparities in high/low volume centre use, but the underlying co-morbidity data isn't presented and should be in table 1 if it is included in the later models.

Thank you for these comments. As noted in response to a comment from Reviewer 1, stage data for the cohort are presented in Supplementary Table 1; stage distribution is broadly comparable between ethnic groups. We have also added the following to the Discussion section regarding stage of disease:

‘Finally, we have focussed on travel required to access cancer treatment within this manuscript; however, in the broader context of addressing disparities in cancer survival for Māori patients, it is also important to examine travel required to access cancer diagnosis. Such an analysis is out of scope for the current study, but future work in this area could include an assessment of ethnic differences in travel distance to screening hubs and secondary diagnostic facilities.’

Regarding data on underlying comorbidity, we have added this data to the manuscript within Supplementary Material 1.

Similarly it would be helpful to discuss if there is any difference in outcomes of liver cancer between the patient groups and if that is associated with the disparities in distance / timing? Presumably even if it cannot be included at patient level in this study there is national cancer outcome data with ethnicity to inform the discussion? I may have missed it but I couldn't see if this was mentioned.

Thank you for this comment. The disparities in cancer survival between Māori and European patients for liver and stomach cancer are the catalyst for the manuscript – as such, in the Introduction we state: 

‘Liver and stomach cancers are among the most important causes of cancer death for the Indigenous Māori population of New Zealand.4 While disparities in the key exposures that drive the incidence of these two cancers is key to the high liver and stomach cancer mortality burden affecting Māori in New Zealand,1,5-7 another important factor is disparities in cancer survival once diagnosed: Māori with liver cancer are nearly a third (31%) more likely to die than non-Māori liver cancer patients, and those with stomach cancer are nearly a quarter (22%) more likely to die than non-Māori stomach cancer patients.8 Identifying and understanding the drivers of these inequities is crucial to informing systemic change aimed at dismantling them.9’ 

For the multivariate models, these should be available in tables with all their included covariates. If not in the main paper these should be in the supplementary information. The tables with the adjusted odds ratios should at least list the included covariates in the table footnotes.

Thank you for this comment. In the context of the current study, we are interested in only the primary modelled relationship between a given distance/time variable and ethnicity (as opposed to other variables, such as age or sex). As such, we believe presenting results for modelled variables other than ethnicity would be confusing, and as such decided not to present them. However we have, as suggested by the review, listed all included covariates within the table footnotes.

 

References

 1. Chamberlain J, Sarfati D, Cunningham R, et al: Incidence and management of hepatocellular carcinoma among Māori and non-Māori New Zealanders. Australian & New Zealand Journal of Public Health 37:520-526, 2013

 2. Signal V, Sarfati D, Cunningham R, et al: Indigenous inequities in the presentation and management of stomach cancer in New Zealand: a country with universal health care coverage. Gastric Cancer 18:571-579, 2015

 3. Gurney J, Stanley J, Jackson C, et al: Stage at diagnosis for Maori cancer patients: disparities, similarities and data limitations. New Zealand Medical Journal 133:43-64, 2020

 4. Gurney J, Robson B, Koea J, et al: The most commonly diagnosed and most common causes of cancer death for Maori New Zealanders. New Zealand Medical Journal 133:77-96, 2020

 5. Signal V, Gurney J, Inns S, et al: Helicobacter pylori, stomach cancer and its prevention in New Zealand. Journal of the Royal Society of New Zealand 50:397-417, 2020

 6. Teng AM, Blakely T, Baker MG, et al: The contribution of Helicobacter pylori to excess gastric cancer in Indigenous and Pacific men: a birth cohort estimate. Gastric Cancer 20:752-755, 2017

 7. Blakely T, Bates MN, Baker MG, et al: Hepatitis B carriage explains the excess rate of hepatocellular carcinoma for Maori, Pacific Island and Asian people compared to Europeans in New Zealand. International Journal of Epidemiology 28:204-210, 1999

 8. Gurney J, Stanley J, McLeod M, et al: Disparities in Cancer-Specific Survival Between Māori and Non-Māori New Zealanders, 2007-2016. JCO Global Oncology 6:766-774, 2020

 9. Gurney J, Campbell S, Jackson C, et al: Equity by 2030: achieving equity in survival for Maori cancer patients. New Zealand Medical Journal 132:66-76, 2019

---

## [Editor Report · Decision Letter 1]

25 May 2022

Equity of travel required to access first definitive surgery for liver or stomach cancer in New Zealand

PONE-D-21-38090R1

Dear Dr. Gurney,

We’re pleased to inform you that your manuscript has been judged scientifically suitable for publication and will be formally accepted for publication once it meets all outstanding technical requirements.

Kind regards,

Aloysious D Aravinthan, MBBS, FRCP, PhD

Academic Editor

PLOS ONE

---

## [Editor Report · Acceptance letter]

4 Aug 2022

PONE-D-21-38090R1 

Equity of travel required to access first definitive surgery for liver or stomach cancer in New Zealand 

Dear Dr. Gurney:

I'm pleased to inform you that your manuscript has been deemed suitable for publication in PLOS ONE. Congratulations! Your manuscript is now with our production department. 

Kind regards, 

on behalf of

Dr. Aloysious Dominic Aravinthan 

Academic Editor

PLOS ONE